# The Impact of the COVID-19 Pandemic on School-Aged Children with Fragile X Syndrome

**DOI:** 10.3390/genes13091666

**Published:** 2022-09-17

**Authors:** Hailey Silver, Hilary Rosselot, Rebecca Shaffer, Reymundo Lozano

**Affiliations:** 1Department of Counseling and Clinical Psychology, Teachers College, Columbia University, New York, NY 10027, USA; 2National Fragile X Foundation, McLean, VA 22102, USA; 3Department of Pediatrics, Cincinnati Children’s Hospital Medical Center, University of Cincinnati College of Medicine, Cincinnati, OH 45229, USA; 4Department of Genetics and Genomic Sciences, Icahn School of Medicine at Mount Sinai, New York, NY 10029, USA; 5Department of Pediatrics, Icahn School of Medicine at Mount Sinai, New York, NY 10029, USA

**Keywords:** Fragile X Syndrome, caregiver perspectives, COVID-19 pandemic

## Abstract

The pandemic caused by the spread of the coronavirus disease (COVID-19), beginning in early 2020, had an impact beyond anything experienced in recent history. People with Fragile X Syndrome (FXS), the leading known heritable cause of ASD and intellectual disability, were uniquely vulnerable to pandemic-related changes. This study surveyed parent perspectives of the impact on 33 school-aged children with FXS across daily living skills, education, therapies, behaviors, health visits, and mask wearing. Academic performance was perceived to have decreased in most of the children (58%). Students in online school had the most reports of decline and those in person had the most reported improvement. Parents were significantly more satisfied with services that remained in person compared to those delivered online or in hybrid settings. Additionally, depression (75%), sleep problems (80%), attention problems (73%), and social skills (61%) were reported to have worsened the most. Parents reported that in addition to continuing with a structured schedule, the most helpful strategies were increasing face-to-face social interactions and outdoor activities. Future research should explore strategies to help online interventions and education to be more successful with individuals with FXS, given this may become a resource for families not geographically able to access in-person resources.

## 1. Introduction

### 1.1. Background

The pandemic caused by the rapid spread of the coronavirus disease (COVID-19), beginning in early 2020, had an impact beyond anything experienced in recent history. Beyond its properties as a disease itself, the pandemic affected people’s lives as they quarantined, social distanced, and took other measures to decrease the spread of the disease. This had a negative impact on the mental health of school-aged children [1,2]. Significant increases were observed in the prevalence of disorders, such as anxiety, depression, distress, and related symptoms cross culturally [3], and there was a profound impact on children’s lifestyles, mental health, and anxiety [2].

Fragile X Syndrome (FXS) is the leading known monogenetic cause of autism spectrum disorder (ASD) and intellectual and developmental disabilities (IDDs). The phenotype of FXS can be described as causing deficits in behavioral, cognitive, adaptive, and emotional functioning. While the mental health and wellbeing of all school-aged children is paramount, it is especially important to understand the impact of this novel pandemic on children with disabilities, such as FXS. Only one prior study has specifically investigated the pandemic impact on FXS [4]. They found perceived increases in behavioral problems and sleep disturbances in children with FXS. These sleep disturbances were thought to be the result of the changes in daily routines that occurred due to the lockdown. However, they found no differences in emotional or prosocial behaviors during lockdown when compared to reports of those behaviors prior to lockdown.

Some information can be inferred about the pandemic’s impact on FXS based on the research that was conducted on ASD and IDDs. In families of children with ASD and IDDs, parent stress was found to have increased due to barriers to receiving services [5,6], taking on tasks, such as teaching or conducting therapies [6,7], and thoughts of possible regressions in behaviors and skills due to decreases or pauses in services [6,8]. Specifically, social skills were reported to have decreased in children with ASD and IDDs [5] and the severity and frequency of behavior problems were reported to have increased [9]. Another study reported improvements in cognitive and language skills in children with ASD since the onset of the pandemic [5].

### 1.2. Study Objective

While ASD and IDD within FXS are similar in many ways to idiopathic forms of ASD and IDD, there may be particular differences in domains, such as social skill functioning, anxiety, and perseveration. Therefore, this study aimed to explore the effects of the COVID-19 pandemic on school-aged children with FXS based on parent-reported changes in behaviors, adaptive skills, and daily activities, such as education and specialized services.

## 2. Materials and Methods

We distributed an online survey to parents of school-aged children with FXS. This survey covered a broad range of topics in order to understand the impact of the COVID-19 pandemic. This study was examined by the Teachers College at Columbia University Institutional Review Board (ID: 21-044) and was considered to be exempt and did not need approval. A recruitment link for the survey was electronically distributed by the National Fragile X Foundation (NFXF) via email. Data were collected from November 2020 through to January 2021 and results represent the pandemic during that time.

The survey was completed by parents or legal guardians of school-aged children with a diagnosis of FXS with a full mutation. More specifically, the child for whom the parent/guardian was completing the survey on behalf of had to be eligible to receive school services for the calendar year in which this survey covered: September 2020 to June 2021. For children who are receiving special education services, this may include any youth between the ages of 3 and 21 years of age.

This study used data collected by an anonymous, voluntary survey. There were no research-validated instruments that were appropriate to collect data on this topic and for this population; therefore, a novel survey was developed (the survey is available in Appendix A). A survey from recent literature with similar topics [9] was used to create the structure of the survey and to gain inspiration on possible domains to investigate. The survey was approved and finalized by experts on FXS through the NFXF’s Research Readiness Program which consisted of a group of multidisciplinary medical doctors (geneticist, pediatrics, neurology, and psychiatry), psychologists, and parents, all with extensive experience related to FXS. The survey sections include demographics, school settings, services/therapies, behaviors, medications, health visits, daily living skills, mask use, and free response. There were two to four questions in each section, for a total of 27 questions, which consisted of multiple choice, Likert-scale, and free-response questions.

Descriptive statistics and other analyses were completed via Microsoft Excel for Mac Version 16.64, sourced from New York, USA. Most results were based upon descriptive statistics. The difference between the time in services before and during the pandemic was analyzed using a 2-tailed, paired *t*-test at a 0.05 significance level. In all other cases, categorical comparisons were calculated in Microsoft Excel by comparing the proportion of responses in each category between groups. The categories were created based upon answers to Likert-scale questions.

## 3. Results

Thirty-three parents of school-aged individuals with FXS (*n* = 33) completed the online survey. The demographics of the individuals with FXS represented are in Table 1. The individuals with FXS represented in the sample ranged in age from 3 to 20 years ($\bar{X}$ = 12.3 years) and were mostly male (*n* = 27, 82%), White (*n* = 30, 91%), and from suburban areas (*n* = 19, 58%), with a wide range of family incomes. The average familial annual income fell between USD 100,000 and USD 150,000.

The respondents were placed in various educational settings prior to the pandemic, as represented in Table 2. Parents were asked to indicate the overall change in their child’s academic performance since the start of the pandemic on a Likert scale from 1, “significantly decreased”, to 5, “significantly increased”. The average ($\bar{X}$ = 2.5, *s* = 1.15) fell within the zone of “mildly decreased,” and there were few responses indicating perceived academic improvement (*n* = 6, 18%).

When using means to investigate the relation between change in academic performance and the method of education during the pandemic, the children who remained in person had the most perceived improvements in academia ($\bar{X}$ = 3.1, *s* = 0.90), those who moved to online learning had the most perceived decreases in academic performance ($\bar{X}$ = 1.9, *s* = 0.83), and the reports of children who moved to hybrid settings ($\bar{X}$ = 2.5, *s* = 1.30) reflected the average of the total sample ($\bar{X}$ = 2.5, *s* = 1.15).

### 3.1. Services and Therapies

The list of services parents reported their children with FXS obtained is in Table 2. Data were analyzed using responses from parents of children who received each service, with parents responding per service. By comparing the proportion of responses of “helpful” to that of the responses of “unhelpful” and neutral combined (Figure 1), parents reported the most helpful services were tutoring (*n* = 7, 70%), speech and language therapy (*n* = 19, 68%), sensory integration therapy (*n* = 2, 67%), and occupational therapy (*n* = 13, 65%). The perceived least helpful services were physical therapy (*n* = 4, 36%), vocational training (*n* = 4, 36%), and counseling (*n* = 1, 14%). When comparing parent reports for services based on their delivery method mid-pandemic, in-person services had the highest rate of satisfaction (*n* = 44, 76%), followed by hybrid (*n* = 11, 46%), and then online services (*n* = 16, 40%). Satisfaction with hybrid and online services was not significantly different. All services decreased in frequency per week, except for vocational training, which increased. However, none of the changes in frequency were found to be statistically significant.

Overall, the mean response of parent perception to how their child adjusted to obtaining services online ($\bar{X}$ = 2.6, *s* = 1.22) fell between “poorly” and “acceptably”. Most people chose “acceptably” (*n* = 8, 29%) and the fewest chose “very well” (*n* = 2, 7%).

### 3.2. Behavior

Parents reported their child’s behavior change from prior to the pandemic to the time of response. The reported improvements in behavior were compared against the reported worsening of behavior for each (Figure 2). Responses of “no change” were not included in these calculations. The highest proportions of improvement were reported in self injury (*n* = 4, 40%) and anxiety (*n* = 9, 32%). However, both of these percentages were still below half of the children who were reported to have each behavior. The highest proportions of worsening were reported in sleep problems (*n* = 12, 80%), depression (*n* = 3, 75%), and attention problems (*n* = 24, 73%).

To understand the impact of the school method on behavior changes during the pandemic, the relationship between parent-reported changes in behavior and the school method mid-pandemic was analyzed. Using a categorical comparison, students who remained in person, compared to other school methods, had the least reported behavior worsening in six of the ten behaviors: aggression (*n* = 2, 50%), attention problems (*n* = 3, 43%), hyperactivity (*n* = 3, 43%), anxiety (*n* = 3, 42.9%), irritability (*n* = 2, 40%), and hypersensitivity (*n* = 1, 17%). Compared to children in other educational methods, children who participated in fully online learning showed more improvement in four behaviors: attention problems (*n* = 2, 18%), hyperactivity (*n* = 2, 18%), aggression (*n* = 2, 25%), and irritability (*n* = 4, 40%).

### 3.3. Daily Living Skills

Parents reported their child’s daily living skill change from prior to the pandemic to the time of response. The mean overall reported change in daily living skills ($\bar{X}$ = 2.6, *s* = 1.14) fell between “mildly decreased” and “stayed the same”. Parents also reported changes in each of five skills (Figure 3). Parents reported the most significant “decline” in social skills (*n* = 20, 61%) and reported the most significant “improvement” in feeding (*n* = 8, 24%). The skills with the least reports of “decline” were bathing (*n* = 5, 15%) and toileting (*n* = 7, 12%). However, a majority of parents reported no change across all five social skills.

Changes in daily living skills or adaptive skills are also important to understand in terms of their interaction with school methods during the pandemic. Students who remained in person for school had the lowest frequency of overall skill decline compared to the other groups (*n* = 3, 43%) and the highest frequency of improvements in overall daily living skills compared to other groups (*n* = 3, 43%). Additionally, students who moved to online instruction had the highest frequency of overall skill declines compared to the other groups (*n* = 6, 55%) and the lowest frequency of improvements in overall daily living skills compared to other groups (*n* = 1, 9%). Overall, social skills had the most significant reports of worsening across all school methods. The reported worsening of social skills was least prevalent in students who remained in person (*n* = 3, 43%) and most prevalent in students who were in school online (*n* = 8, 73%). Both feeding (*n* = 4, 36%) and toileting (*n* = 4, 36%) had a spike in reported behavior worsening amongst students who were in online education settings compared to students in hybrid and in-person settings. Students who remained in person had the highest prevalence of reported skill improvement across almost all skills: feeding (*n* = 3, 43%), dressing (*n* = 2, 29%), toileting (*n* = 2, 29%), and social skills (*n* = 2, 29%). Of note, bathing skills had the most consistency in skill change across all three groups and were the only adaptive behavior that was not highest in students who remained in person for school during the pandemic.

### 3.4. Medication, Health Visits, and Masks

Most parents responded that their children with FXS did not have any medication changes (*n* = 29, 88%). However, most respondents had not had appointments with an FXS specialist during the period since the pandemic shutdown (*n* = 23, 70%). Of those that had seen specialists, most visits were in person (*n* = 6, 60%). The appointment type parents were most likely to attend in person was for vaccinations ($\bar{X}$ = 4.2, *s* = 1.16) and annual physicals ($\bar{X}$ = 4.0, *s* = 1.17). The appointments parents were most likely to attend virtually were for medical advice ($\bar{X}$ = 3.9, *s* = 1.24), behavioral changes ($\bar{X}$ = 3.7, *s* = 1.29), and regular follow-ups ($\bar{X}$ = 3.7, *s* = 1.36). Overall, positive reports of ability to wear a mask (*n* = 21, 64%) outweighed negative reports (*n* = 12, 36%). The most common challenge reported was that students with FXS “do not understand the necessity of a mask” (*n* = 15, 46%).

### 3.5. Free Response

Parents were allowed to expand upon any additional comments they felt were important to report. Parents seemed to observe notable changes in their child’s anxiety both increasing and decreasing during the pandemic. Parents also reported that they need more support than what they are currently obtaining during the pandemic, a common theme in past literature as well [4,5,6]. Parents noted that virtual therapies were difficult to navigate and observed significant regressions in their child’s social skills, especially with the difference in social skills necessary for virtual platforms. Parents recommend implementing visual daily schedules (*n* = 11, 33%), increasing face-to-face activities and social interactions (*n* = 5, 15%), staying active and going outdoors (*n* = 4, 12%), and increasing communication from teachers (*n* = 3, 9%).

## 4. Discussion

This study aimed to understand the impact of the COVID-19 pandemic on multiple domains in school-aged children with FXS. Children with FXS faced more challenges during the pandemic. Social skills were the most negatively impacted factor by the pandemic. Other negatively impacted behaviors were sleep problems, depression, and attention problems. Despite prior conclusions that anxiety, depression, and related mental health disorders had large prevalence increases in neurotypical school-aged children [3], this was not necessarily the case in school-aged children with FXS. While depression was one of the behaviors that was reported to have worsened the most, anxiety and self injury showed the most improvement. However, parent’s anecdotal responses in the free response section differed in their perspective of their children’s change in anxiety. They indicated both serious increases and decreases in their child’s anxiety during the pandemic. This may reflect the clinical variability in FXS and the different contexts where anxiety is exacerbated. Further research should be conducted to investigate possible mediating factors in this change, such as time spent in familiar settings/comfort zones or sudden changes in routine and environment.

Services that continued in person were considered the most beneficial by parents and guardians of children with FXS and online settings were reported to be the least beneficial for therapies and services. Additionally, there were no significant changes in service frequency found when calculated by therapy/service. The services reported to be most beneficial during the pandemic were tutoring, sensory integration therapy, speech language therapy, and occupational therapy and those reported to be least beneficial were physical therapy, vocational training, and counseling. These may have been reported to be less beneficial because of their important components that could have been more easily disrupted by the pandemic shifts in social distancing and in-person opportunities.

Children who remained in person for school had a higher average academic improvement than children who switched to online or hybrid school settings, suggesting that students with FXS benefit the most, academically, from in-person instruction. Students who remained in person for school had the least reported decline in most behaviors, with rates of worsening attention problems and hyperactivity being significantly lower in students who remained in person when compared to students who moved to hybrid or online settings. Students who shifted to online learning had the most significant improvements in irritability. Further research should be conducted to understand why there is such a relationship between these behaviors and educational methods.

Overall daily living skill changes were reported to be most improved and least declined in children who remained in person and they were most declined and least improved in children who were moved to entirely online education. While social skill decline was evident across all education methods, the least decline in this skill was in children who remained in person and the greatest decline was in children who received online education. Children who were in school online also had a larger decline in feeding and toileting skills compared to children who were in other schooling methods, whereas children who remained in person had the most improvements in all but one daily living skill: bathing; this remained constant across all methods.

Based on these results, the perceived benefit of keeping school-aged children with FXS in their daily routines and remaining in person for school and related services is clear. While online learning has been found to have similar academic effects to in-person learning for neurotypical children [10], this did not seem to be the case for children with FXS based on their parent’s reports. While results from this survey were able to provide a general understanding of the trends based on the impact of the pandemic on this population, this is hopefully only the beginning for studies on the pandemic impact for this and similar populations and limitations to these results provide clear opportunities for further exploration.

It is important to consider the clinical variability in FXS. There may be sub-phenotypes of FXS, which would account for the variability in responses to the pandemic and its impact on the domains surveyed. This variability in FXS phenotypes may account for differences in adjustment to the COVID-19 pandemic and future research should be conducted to analyze the interaction between sub-phenotypes and the impact of the COVID-19 pandemic.

### Limitations

Limitations of the study include the small sample and that this was a cross-sectional survey. Survey respondents were asked to reflect on their child’s behavior prior to the pandemic; therefore, recall bias may have occurred. Additionally, this survey collected parent and guardian opinions of their child’s development throughout the pandemic, which could also have added bias. The parents or caregivers who chose to respond to this survey may also present a limitation, since they may reflect a bias of opinions of the pandemic’s impact on their children, skewing the results either positively or negatively compared to the overall population. This study only showed correlation between time points related to the pandemic and, therefore, did not measure direct causation. Finally, this study used a survey format rather than a validated assessment instrument due to the lack of appropriate instruments present to measure the desired domains.

## 5. Conclusions

Survey responses indicated that, more than anything, parents were disappointed in shifts to online education, therapies, and services. This disappointment was substantiated by data based on respondent observations of their children’s changes in behaviors and daily living skills since the onset of the pandemic, because both domains indicated a higher rate of worsening in children who had moved to online education as compared to children who remained in person for school. Overall, some of the largest perceived declines in behaviors and skills were in social skills, depression, and sleep problems. Parents also noted both substantial increases and substantial decreases in their child’s anxiety.

While this study created a foundation for understanding the impact of the pandemic on school-aged children with FXS, there is a lot of future research that is possible on this topic. The data obtained in this study and other future studies can be used to inform clinicians, teachers, and specialists who work with children with FXS and afford them the opportunity to better understand the unique impact this pandemic has had on these children and, furthermore, improve their ability to target the main deficits and challenges that have faced this population since the onset of the pandemic. Future research should also explore strategies to help online interventions and education to be more successful with individuals with FXS, given this may become a resource for families not geographically able to access in-person resources.

## Figures and Tables

**Figure 1 genes-13-01666-f001:**
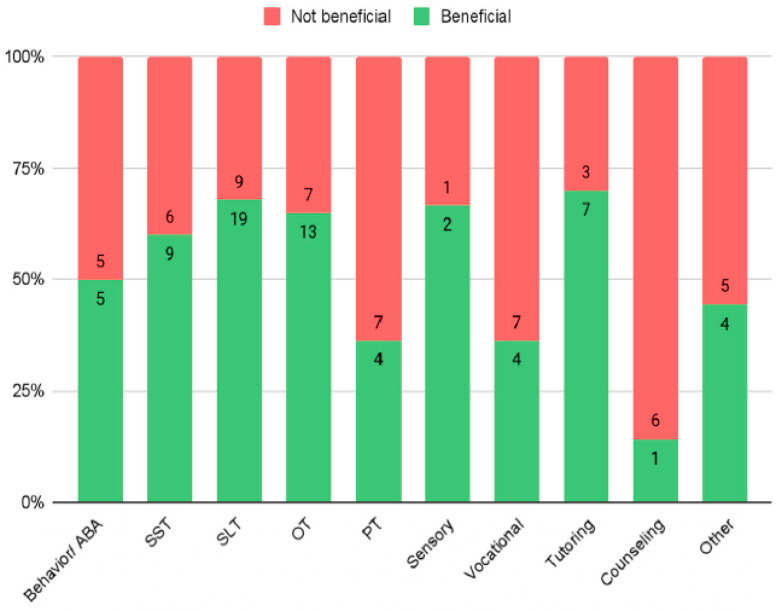
Proportion of how helpful each service was rated.

**Figure 2 genes-13-01666-f002:**
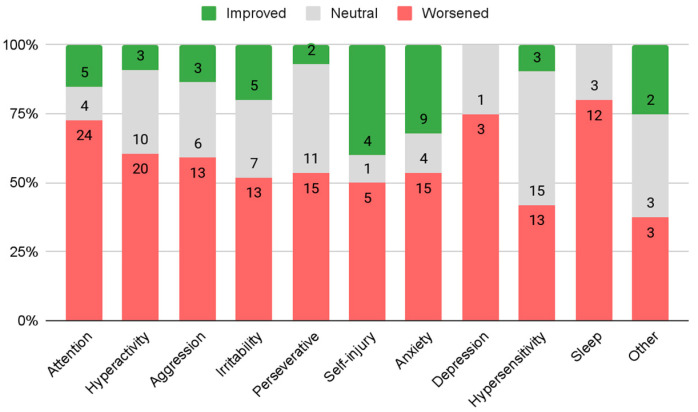
Reported change in behaviors.

**Figure 3 genes-13-01666-f003:**
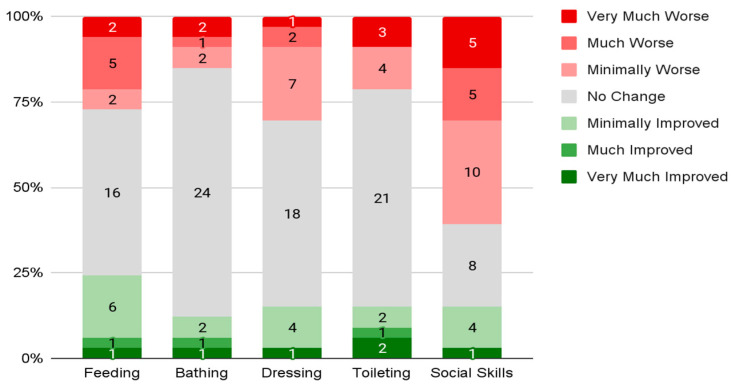
Reported change in daily living skills.

**Table 1 genes-13-01666-t001:** Demographic information for children with FXS represented.

	Total (*n*)	%
**Sex**		
Male	27	82%
Female	6	18%
**Race**		
White	30	91%
Hispanic	2	6%
Asian	1	3%
**Location**		
Urban	6	18%
Suburban	19	58%
Rural	8	24%
**Income**		
<USD 50,000	2	6%
USD 50,000–USD 100,000	9	27%
USD 100,000–USD 150,000	6	18%
USD 150,000–USD 200,000	6	18%
USD > 200,000+	6	18%
Prefer not to answer	4	12%
**Age**		
3–5 years	2	6%
6–9 years	8	24%
10–13 years	12	36%
14–16 years	2	6%
17–20 years	11	33%

**Table 2 genes-13-01666-t002:** Education and services obtained.

	Pre-Pandemic		Mid-Pandemic	
	**Total (*n*)**	**%**	**Total (*n*)**	**%**
**Educational Setting**				
School for children with special needs	9	27%	4	12%
Self-contained classroom, no time in typical classroom	4	12%	2	6%
Self-contained classroom, some time with typical peers	6	18%	2	6%
Typical classroom with classroom support	12	36%	5	15%
Typical classroom without support	1	3%	0	0%
Homeschooling	0	0%	0	0%
Online/cyber schooling	0	0%	12	36%
No schooling, therapy only	1	3%	1	3%
No schooling or therapy	0	0%	0	0%
Other	0	0%	7	21%
	**Total (*n*)**		**%**	
**Teaching method**				
In-person	7		21%	
Hybrid	15		46%	
Online	11		33%	
**Services Obtained**				
Psychological/Behavioral	10		30%	
Social Skills Training	15		46%	
Speech and Language Therapy	28		85%	
Occupational Therapy	20		61%	
Physical Therapy	11		33%	
Sensory Integration Therapy	2		6%	
Vocational Training	11		33%	
Tutoring	9		27%	
Counseling	6		18%	
Other	9		27%	

## Data Availability

Data from the full survey are available upon request.

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
