# Peer review of "The Impact of the COVID-19 Pandemic on School-Aged Children with Fragile X Syndrome"

_genes, 2022, doi:10.3390/genes13091666_

Round 1

Reviewer 1 Report

The present article deals with an increasingly topical issue. The authors focus on the specific limits of the pandemic in individuals suffering from Fragile X Syndrome. In general, I find the topic interesting and suitable for publication in the journal. From a general point of view, it can be assessed as original. However, I have the following comments on its proper publication:

1. Introduction: this is a pivotal chapter to provide insight into the chosen topic. Even given the small number of sources, I can imagine this chapter being better developed. At the very least, it would be useful to add 1.1 Background. In this subchapter the limits and problems will already be clear. There will no longer be general descriptions in this subchapter as there are now. Please supplement this subchapter with additional sources, after all, having only 10 sources is insufficient for a scientific paper. From this subchapter, it is appropriate to conclude by stating the objectives of the research investigation that has been carried out. This will be smoothly followed by chapter 2. 

Chapter 2, Materials and Methods, offers quite good information on the data processing and presentation sequence. Here I have no major objections. Only perhaps it would be useful to add what statistical procedure (program) was chosen. Maybe I am just not familiar with it and it is already in the text, but I would have welcomed it in this chapter.

3. Results: no problem. The presentation is appropriately implemented with respect to the data obtained.

4. Discussion: in the discussion I would have liked more confrontation with other sources. It would be appropriate to add e.g. the possibility of comprehensive rehabilitation approaches that can contribute to improving the educational process even during pandemics or outside of pandemics. I recommend to add to the discussion general information about the possibilities of holistic rehabilitation, as holistic approaches of different helping professions (I recommend to focus e.g. on this article, which I first found in the MDPI archive, https://www.webofscience.com/wos/woscc/full-record/WOS:000794373400001), and this article (https://www.webofscience.com/wos/woscc/full-record/WOS:000545292800010) can also be used to support this.

The point is that the discussion is also meant to provide a broader perspective and options on the issue, here the authors still stick to a specific topic and that is a pity. The article provides interesting information, it just needs to fine tune the individual chapters and the presentation itself.

5. Chapter conclusion: I recommend adding limits to the study. What might have limited the results obtained? Please add.

Overall, however, the article gives a good impression.

Author Response

Response to Reviewer 1 Comments

Point 1: Introduction: this is a pivotal chapter to provide insight into the chosen topic. Even given the small number of sources, I can imagine this chapter being better developed. At the very least, it would be useful to add 1.1 Background. In this subchapter the limits and problems will already be clear. There will no longer be general descriptions in this subchapter as there are now. Please supplement this subchapter with additional sources, after all, having only 10 sources is insufficient for a scientific paper. From this subchapter, it is appropriate to conclude by stating the objectives of the research investigation that has been carried out. This will be smoothly followed by chapter 2. 

Response 1: Thank you for your feedback. We added in headings to separate the background section of the introduction from the current study objectives and make this flow clearer to readers. We kept this section simple because there was only one prior study on this topic performed with Fragile X Syndrome and we felt it was not always applicable to cite sources which targeted general ASD and IDDs and allow the reader to make assumptions based on those findings. Additionally, this is a new topic of research as it has only occurred recently. This is why there are only 10 sources and why this section is simpler than most scientific papers.

Point 2: Chapter 2, Materials and Methods, offers quite good information on the data processing and presentation sequence. Here I have no major objections. Only perhaps it would be useful to add what statistical procedure (program) was chosen. Maybe I am just not familiar with it and it is already in the text, but I would have welcomed it in this chapter.

Response 2: One edit was made to make sure this was clear. Thank you for this comment. This information can be found in the last paragraph of section 2, starting on line 108. This study only utilized descriptive statistics and a paired t-test, which were performed in Microsoft Excel, using the proper formulas.

Point 3: Results: no problem. The presentation is appropriately implemented with respect to the data obtained.

Response 3: No response needed. No changes were made. Thank you.

Point 4: Discussion: in the discussion I would have liked more confrontation with other sources. It would be appropriate to add e.g. the possibility of comprehensive rehabilitation approaches that can contribute to improving the educational process even during pandemics or outside of pandemics. I recommend to add to the discussion general information about the possibilities of holistic rehabilitation, as holistic approaches of different helping professions (I recommend to focus e.g. on this article, which I first found in the MDPI archive, https://www.webofscience.com/wos/woscc/full-record/WOS:000794373400001), and this article (https://www.webofscience.com/wos/woscc/full-record/WOS:000545292800010) can also be used to support this.

The point is that the discussion is also meant to provide a broader perspective and options on the issue, here the authors still stick to a specific topic and that is a pity. The article provides interesting information, it just needs to fine tune the individual chapters and the presentation itself.

Response 4: Thank you for your feedback. The purpose of this article is not to postulate on rehabilitation techniques, but rather to point out where the deficits may lie in this population and understand where parents notice regressions or improvement in their children with FXS so that clinicians may find the best ways to help support people with FXS as we come out of this pandemic era or in the case of another pandemic. This hope was stated in the conclusion section on lines 317-321. Not enough information was obtained from this study in order to appropriately ascertain what interventions would be helpful for people with FXS, besides stating parent disappointment with the presentation of services online rather than in person. Further, neither of the articles provided in the critique were tested in FXS, only in stroke victims, cerebral palsy, and developmental dyspraxia, and I do not feel comfortable recommending a treatment for children with FXS based on research studies that were done in populations that are not similar to FXS. Additionally, we do not have enough research on this subject to say that people with FXS made enough regressions to require rehabilitative therapy since this research was based on parent perspectives rather than clinical comparisons of behaviors and skills over time. I believe the purpose of this article is to present information to professionals in this field on a specific topic and going off of that topic would be irrelevant to the study and purpose.

Point 5: Chapter conclusion: I recommend adding limits to the study. What might have limited the results obtained? Please add.

Response 5: Thank you for this suggestion. The limitations can be found in the discussion section rather than the conclusion section. I have added a heading for this section to make their presence clearer. This section starts on line 293. My fellow authors and I felt it was more appropriate to include the limitations in the discussion section rather than the conclusion section.

Thank you for your review of this article. I hope we successfully addressed your comments.

Reviewer 2 Report

This is an interesting article on a timely topic.

Line 69 says the research was exempt and approved, but if it was exempted then it did not need approval.  You could say it was examined by the review board and found to be exempted and not needing approval.

I think you should indicate how many forms were sent out and what the percentage of submissions were of those that were sent out. This could reflect a bias. Maybe only those parents who had some disappointment were replying.

Author Response

Response to Reviewer 2 Comments

Point 1: Line 69 says the research was exempt and approved, but if it was exempted then it did not need approval. You could say it was examined by the review board and found to be exempted and not needing approval.

Response 1: Thank you for catching this error in language. This change has been made to say that the study was “considered to be exempt and did not need approval.” Please find this change on lines 80-82.

Point 2: I think you should indicate how many forms were sent out and what the percentage of submissions were of those that were sent out. This could reflect a bias. Maybe only those parents who had some disappointment were replying.

Response 2: That is a good point, however, we are unaware of how many people were aware of the survey, since it was linked directly to the NFXF website. However, this limitation was added to the discussion section. Please see this addition starting on lines 299-302.

Thank you so much for your review of this article.